# Enhanced locomotor recovery in mice lacking GlcNAc6ST1 and GlcNAc6ST4 following spinal cord injury

Masayoshi Morozumi[1,2,]*, Tomoya Ozaki[1,]*, Kazuchika Nishitsuji[3,4,]*, Yoshiko Takeda-Uchimura[4],
Akiyuki Matsumoto[1,5], Sadayuki Ito[1,2], Shiro Imagama[2], Naoki Ishiguro[2], Hirokazu Yagi[6,7], Koichi Kato[6,7],
Tomoya O Akama[8], Tomoki Kosugi[9], Shoichi Maruyama[9], Kenji Kadomatsu[1,10], Steven D Rosen[11],
Linda J Noble-Haeusslein[12,13], Kenji Uchimura[1,4]

**Spinal cord injury (SCI) damages neural circuits and triggers pro-inflammatory responses, resulting in locomotor impairment. The carbohydrate sulfotransferases GlcNAc6ST1 and GlcNAc6ST4 modulate the function of blood monocytes and microglia. However, their specific roles and enzymatic relationships in neuro-inflammation and functional recovery after SCI remain unclear. In this study, we demonstrate that mice deficient in both GlcNAc6ST1 and GlcNAc6ST4 (DKO) exhibit improved locomotor recovery compared with mice with a single deficiency. DKO mice exhibit reduced levels of monocytes and activated macrophages/microglia at the injury site alongside increased serotonergic neural fibers, indicating enhanced neural plasticity. RNA sequencing reveals down-regulation of collagen I genes and up-regulation of genes encoding synaptic membrane components in the injured DKO spinal cord. In addition, GALAXY glycomic analysis shows an absence of GlcNAc-6-sulfated *N*-glycans in the DKO spinal cord. These results suggest that GlcNAc6ST1 and GlcNAc6ST4 play complementary roles in promoting detrimental inflammatory responses post-SCI. Targeting these sulfotransferases could offer a novel therapeutic strategy to improve locomotor recovery after SCI.**

## Introduction

Traumatic injury to the adult mammalian spinal cord causes long-term loss of locomotor function. Direct damage to the spinal cord disrupts neural circuits and triggers an early pro-inflammatory response coincident with vascular disruption and intra-parenchymal hemorrhage (Silver et al, 2014; Dias et al, 2018; Sofroniew, 2018). Damaged axons have a low potential for regeneration across the astrocyte/fibrotic scar, which is induced after injury, extending into spared tissue because of a high concentration of inhibitory molecules of axonal regrowth and sprouting (Silver et al, 2014; Fawcett, 2015; Adams & Gallo, 2018; Sofroniew, 2018; Linnerbauer et al, 2020; Zheng & Tuszynski, 2023; O'SheaAo et al, 2024). Chondroitin sulfate (CS) is a sulfated glycan covalently attached to core proteins to form CS proteoglycans in the ECM. CS, seen in a gradient pattern around the lesion scar, plays a major role in the inhibition of axonal regrowth and sprouting and synapse remodeling after injury (Davies et al, 1997; Silver et al, 2014; Dyck & Karimi-Abdolrezaee, 2015; Fawcett, 2015; Tran et al, 2022). Enzymatic degradation of the CS glycan (Radbury et al, 2002; Garcia-Alias et al, 2009; Alilain et al, 2011; Rosenzweig et al, 2019) or modulation of its signaling receptors enhances neural regeneration and functional recovery after spinal cord injury (SCI) (Shen et al, 2009; Fry et al, 2010; Fisher et al, 2011; Dickendesher et al, 2012; Lang et al, 2015; Sakamoto et al, 2019; Milton et al, 2023).

Secondary tissue damage because of complications after the primary impact exacerbates neurological dysfunction in the injured spinal cord (McCreedy et al, 2018). Blood monocytes, the precursors of macrophages, are recruited to the central nervous system (CNS) lesion site (Popovich & Hickey, 2001; Zhang et al, 2011; Herz et al, 2017; Werner et al, 2020; Dorrier et al, 2021). Extravasation of monocytes, their differentiation into macrophages, and their subsequent activation along with resident microglia, all of which are a part of the CD11b-positive myeloid lineage, are required for

[1]Department of Biochemistry, Nagoya University Graduate School of Medicine, Nagoya, Japan    [2]Department of Orthopedic Surgery, Nagoya University Graduate School of Medicine, Nagoya, Japan    [3]Wakayama Medical University, Wakayama, Japan    [4]University Lille, CNRS, UMR 8576-UGSF- Unité de Glycobiologie Structurale et Fonctionnelle, Lille, France    [5]Department of Orthopedic Surgery, Okazaki City Hospital, Aichi, Japan    [6]Graduate School of Pharmaceutical Sciences, Nagoya City University, Nagoya, Japan    [7]Exploratory Research Center on Life and Living Systems (ExCELLS), National Institutes of Natural Sciences, Okazaki, Japan    [8]Department of Pharmacology, Kansai Medical University, Osaka, Japan    [9]Department of Nephrology, Nagoya University Graduate School of Medicine, Nagoya, Japan    [10]Institute for Glyco-core Research (iGCORE), Nagoya University, Nagoya, Japan    [11]Department of Anatomy, University of California, San Francisco, CA, USA    [12]Department of Neurosurgery, Physical Therapy and Rehabilitation Science, University of California, San Francisco, CA, USA    [13]Department of Psychology and Neurology, The Dell Medical School, University of Texas, Austin, TX, USA

Correspondence: kenji.uchimura@univ-lille.fr
Tomoya Ozaki's present address is Nagoya City University Graduate School of Pharmaceutical Sciences, Nagoya, Japan
*Masayoshi Morozumi, Tomoya Ozaki, and Kazuchika Nishitsuji contributed equally to this work

the phagocytotic clearance of noxious debris and healing of the damaged spinal cord (Donnelly & Popovich, 2008; David & Kroner, 2011; London et al, 2013; Russo & McGavern, 2015; Brennan et al, 2022). Whereas over time these processes become beneficial, at early stage after injury, they can exert adverse effects on neuronal repair, including the retraction of injured axons upon direct contact with activated macrophages to dystrophic axon tips and secretion of neurotoxic factors by activated macrophages and microglia present at the lesion center (Popovich et al, 1999; Horn et al, 2008; Evans et al, 2014; Kobayakawa et al, 2019; Dorrier et al, 2021). Polarization of activated macrophages and microglia into their neuroprotective states is desirable in the secondary mechanisms of repair, which lasts over weeks or months (Kigerl et al, 2009; Shechter et al, 2013; Kroner et al, 2014).

Interaction of monocytes with hyaluronan is an important step for their recruitment into the lesion site and for cell differentiation, cell polarization, and tissue repair (Lee-Sayer et al, 2015). Pro-inflammatory cytokine TNF$\alpha$-induced binding of monocytes to hyaluronan correlates with increased carbohydrate sulfation of the CD44 hyaluronan receptor (Freeman et al, 2018) in monocytes (Maiti et al, 1998; Streit et al, 1998; Brown et al, 2001). It has been demonstrated that TNF$\alpha$ up-regulates the expression of sulfation modification enzymes *N*-acetylglucosamine-6-*O*-sulfotransferase (GlcNAc6ST) 1 (encoded by the gene *Chst2*) and GlcNAc6ST4 (encoded by the gene *Chst7*) (Uchimura et al, 2005; Uchimura & Rosen, 2006), leading to increased sulfation of the *N*- and *O*-linked glycans of CD44 and hyaluronan binding in monocytes (Brown et al, 2001; Delcommenne et al, 2002; Tjew et al, 2005). We hypothesized that both GlcNAc6ST1 and GlcNAc6ST4 are involved in the inflammatory response after the initial damage in SCI by increasing the recruitment of monocytes to the injury site and promoting the detrimental differentiation of macrophages/microglia into an activated state. We have shown that deletion of GlcNAc6ST1 results in microglial phenotypes polarized into an anti-inflammatory state (Zhang et al, 2017). However, the contribution of GlcNAc6ST4 to neuroinflammation and functional recovery after SCI is unknown. The enzymatic relationship between GlcNAc6ST1 and GlcNAc6ST4 also remains uncertain. In this study, we show that GlcNAc6ST1 and GlcNAc6ST4 are expressed in the CD11b-positive myeloid lineage cells in the injured murine spinal cord. Here, we provide the first evidence that mice doubly deficient (DKO) in GlcNAc6ST1 and GlcNAc6ST4 show improved locomotor recovery. DKO mice also show reduced levels of monocytes and monocyte-derived macrophages at the lesion site and a low abundance of CD68-positive activated macrophages/microglia in the lesion center after SCI. In addition, the DKO mice show an increase in the level of 5-hydroxytryptamine (5-HT)-positive serotonergic neural fibers and a greater amount of residual white matter in the injured spinal cord. The RNA-sequencing (RNA-Seq) analysis of the transcriptome indicates a down-regulation of genes associated with collagen I and ECM components; alongside, an up-regulation of genes encoding integral components of synaptic membranes is up-regulated in the injured spinal cord of the GlcNAc6ST1 and GlcNAc6ST4 DKO mice. The GALAXY glycomic analysis reveals the absence of GlcNAc-6-sulfated *N*-glycans in the injured DKO spinal cord. These results stand in contrast to those observed in KO models of GlcNAc6ST3 (encoded by the *Chst5*) and KSGal6ST (encoded by the

*Chst1*), where locomotor recovery is comparable to that of spinal cord injured WT mice. Thus, GlcNAc6ST1 and GlcNAc6ST4 are complementarily involved in the pathogenesis of SCI by increasing the recruitment of monocytes and up-regulating the detrimental differentiation of macrophages/microglia at the lesion center. The present study provides a new therapeutic basis for enhancing locomotor recovery after SCI by targeting these two sulfotransferases.

# Results

### GlcNAc6ST1 and GlcNAc6ST4 mRNA expression within the CD11b-positive myeloid cells in the injured spinal cord

We had previously shown that mRNAs of GlcNAc6ST1, GlcNAc6ST3, and GlcNAc6ST4 are expressed in the adult mouse brain (Narentuya et al, 2019). In this study, we analyzed the mRNA expression of GlcNAc6ST family members in the spinal cord of WT mice after injury. Mice were subjected to contusive SCI or a laminectomy surgery at the T10 vertebral level and then sacrificed at 7 d post-operation. In all mice, 3-mm spinal cord segments, both rostral and caudal to the lesion epicenter, were collected (Fig 1A). GlcNAc6ST1, GlcNAc6ST3, and GlcNAc6ST4 expression was detected in all segments. GlcNAc6ST2 expression was negligible in both sham controls and injured cords (the values were either not detected or ≤0.00015). At the lesion center, a 2.4-fold increase in the GlcNAc6ST4 mRNA level was observed 7 d after the injury (Fig 1B). CD11b is a myeloid cell-specific integrin that mediates adhesion, migration, and accumulation of myeloid lineage cells during inflammation. As a marker for monocytes and macrophages/microglia, CD11b was used. mRNA level in the lesion center was 17-fold higher than that in the corresponding segment in the intact cord. A 4- to 7-fold increase in the level of CD11b mRNA was observed in 3-mm-long rostral and caudal segments in injured mice (Fig 1B). To investigate whether GlcNAc6STs are expressed in CD11b-positive myeloid cells in the injured spinal cord, we isolated primary CD11b-positive cell populations by magnetic beads conjugated with an anti-CD11b antibody. GlcNAc6ST1 and GlcNAc6ST4 mRNAs were expressed in primary CD11b-positive cells (Fig 1C). The GlcNAc6ST4-KO allele was replaced by the lacZ gene cassette (Narentuya et al, 2019). To determine the localization of the cells that express lacZ in the GlcNAc6ST4-KO allele, X-gal staining was performed for the spinal cord sections prepared from GlcNAc6ST4-KO allele heterozygotes 7 d after the injury. Cells accumulating at the abluminal side of blood vessels within the lesion center were intensely stained with X-gal (Fig 1D). Single-cell RNA-Seq data (Hamel et al, 2023) were analyzed to determine which CD11b-positive myeloid cells in the injured mouse spinal cord expressed GlcNAc6ST1 and GlcNAc6ST4 mRNAs. As expected, mRNA expressions of GlcNAc6ST1 and GlcNAc6ST4 were observed in both infiltrating monocyte-derived macrophages and microglia (Fig S1). These genes were also expressed in dendritic cells. Both GlcNAc6ST2 and GlcNAc6ST3 were at negligible levels in the myeloid lineage cells in the injured mouse spinal cord.

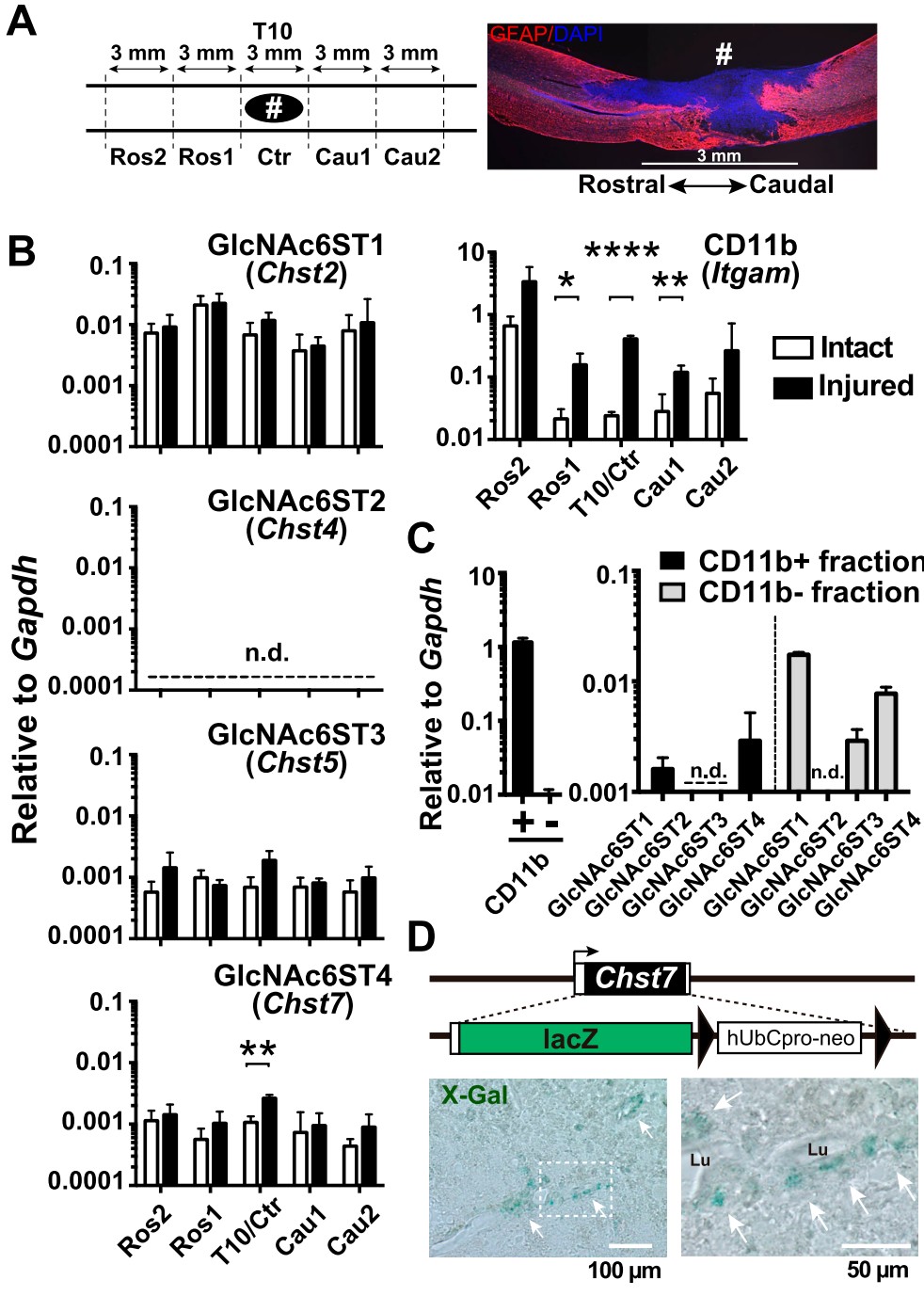

**Figure 1. mRNA expression of *N*-acetylglucosamine-6-*O*-sulfotransferase (GlcNAc6ST) 1 and GlcNAc6ST4 in myeloid lineage cells of spinal cord after a contusive injury.**

**(A)** A contusive spinal cord injury was produced using a 100k-dyn force at the T10 vertebral level of adult mice. The diagram shows preparation of 3-mm spinal cord segments rostral (Ros1, Ros2) and caudal (Cau1, Cau2) to the lesion center (Ctr). Sections of the injured spinal cord were subjected to immunostaining using an antibody specific to glial fibrillary acidic protein (GFAP), a marker for astrocytes (*red*), and DAPI for nuclear staining (*blue*). A representative longitudinal confocal image of the injured spinal cord of a WT mouse is shown. The position of the epicenter (the region of maximal damage) is shown (#). **(B)** mRNA expression levels of GlcNAc6STs and CD11b, a myeloid lineage marker, in the spinal cord 7 d post-injury were quantified using qRT-PCR (*n* = 4 for sham control group [intact]; *n* = 3 for injured group). Data are means ± s.d. *$P < 0.05$, **$P < 0.01$, ****$P < 0.0001$. n.d., not detected. **(C)** CD11b+ macrophages/microglia and monocytes were isolated from the spinal cord 7 d post-injury using a MACS microglial isolation kit (Miltenyi Biotec). mRNA levels of GlcNAc6ST were assessed in CD11b bead-bound (CD11b+) or unbound (CD11b−) cells (*n* = 3 per group). **(D)** Schematic diagram showing the locus of GlcNAc6ST4/Chst7 and a homologous recombination construct with the cassette of the *E. coli* lacZ gene (lacZ) and the human ubiquitin C gene promoter (hUbCpro)/neomycin resistant gene (neo) flanked by loxP. X-gal staining was performed for a spinal cord of a GlcNAc6ST4 heterozygous mouse 7 d post-injury. X-gal staining was intense for the cells aggregating in proximity of blood vessels in the abluminal area (*white arrows*). Lu, lumina of blood vessels in the lesion center. Representative bright-field images of a transverse section are shown. Source data are available for this figure.

## GlcNAc6ST1 and GlcNAc6ST4 double deficiency improves locomotor recovery after SCI

Considering these results, we wanted to investigate whether GlcNAc6ST1 and GlcNAc6ST4 are involved in locomotor recovery after SCI, in particular by regulating the functions of myeloid lineage cells. The Basso mouse scale (BMS) locomotor rating scale (Basso et al, 2006), which is an open-field test paradigm, was used to assess the locomotive outcomes in a double-blinded manner.

Repeated measures two-way ANOVA showed that postoperative time and GlcNAc6ST deficiency had significant effects on the BMS locomotor function recovery (Fig 2). Post hoc analysis using Tukey's range test revealed that mice single deficient in either GlcNAc6ST1 or GlcNAc6ST4 displayed locomotive function comparable with WT control mice after injury (Fig 2). Interestingly, GlcNAc6ST1 and GlcNAc6ST4 doubly deficient (DKO) mice showed a significant improvement in locomotive function starting at 4 wk post-injury and weekly thereafter until 8 wk post-injury (versus WT

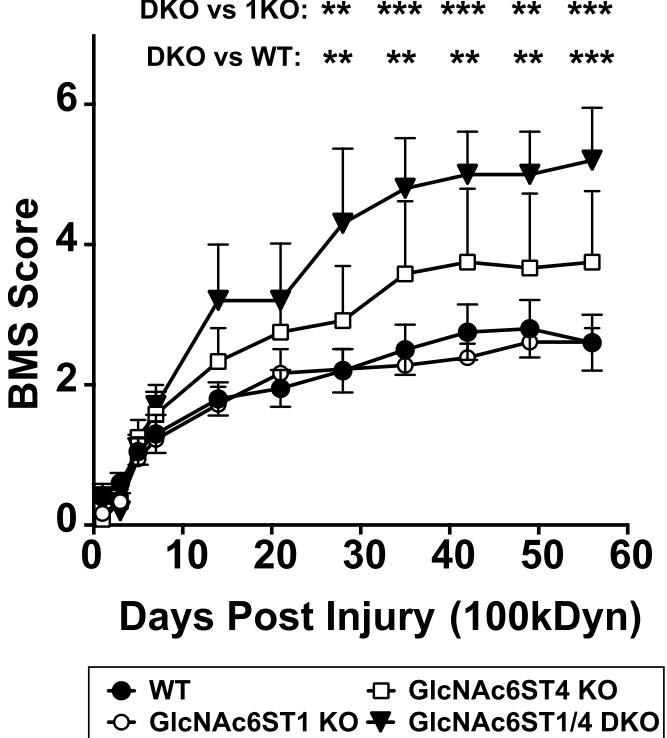

DKO vs 1KO: **  ***  ***  **  ***
DKO vs WT: **  **  **  **  ***

**Figure 2. Functional locomotor recovery after contusive spinal cord injury in GlcNAc6ST1 and GlcNAc6ST4 double deficient mice.**
Locomotor recovery was evaluated using the Basso mouse scale (BMS) (Basso et al, 2006) over 8 wk post-injury in a blinded manner ($n$ = 10 for WT; $n$ = 9 for GlcNAc6ST1-KO; $n$ = 6 for GlcNAc6ST4-KO; $n$ = 5 for GlcNAc6ST1/4 DKO). Repeated measures two-way ANOVA with days post-injury and genotype as factors revealed significant effects on locomotor recovery (days post-injury: $F_{10, 260}$ = 71.00, $P$ < 0.0001; genotype: $F_{3, 26}$ = 3.111, $P$ < 0.05; interaction between days post-injury and genotype: $F_{30, 260}$ = 2.822, $P$ < 0.0001). Post hoc Tukey's range test showed significant locomotor recovery, starting at 4 wk post-injury and weekly thereafter until 8 wk post-injury in GlcNAc6ST1/4 DKO mice compared with WT or GlcNAc6ST1-KO. Data are means ± s.e.m. **$P$ < 0.01, ***$P$ < 0.001. Source data are available for this figure.

or GlcNAc6ST1-KO) (Fig 2). We next examined if loss of sulfation modifications, namely, GlcNAc-6-sulfation and galactose (Gal)-6-sulfation, in the neuropil keratan sulfate polysaccharide improve the locomotor recovery after SCI. We have previously reported that GlcNAc6ST3 is a major GlcNAc6ST of the neuropil keratan sulfate polysaccharide expressed in oligodendrocytes (Narentuya et al, 2019) and that KSGal6ST is a major Gal-6-$O$-sulfotransferase of the keratan sulfate polysaccharide (Hoshino et al, 2014). KSGal6ST was expressed in both intact and injured spinal cord (Fig S3A). Repeated measures two-way ANOVA showed that mice deficient in GlcNAc6ST3 or KSGal6ST had no improvement in locomotive outcomes after SCI, compared with WT mice (Figs S2 and S3B). Mice deficient in C6ST1 (encoded by the $Chst3$), which has dual Gal-6-$O$-sulfotransferase and $N$-acetylgalactosamine (GalNAc)-6-$O$-sulfotransferase activities, also showed comparable locomotor recovery after SCI (Fig S3B), consistent with the findings of a previous report that the lack of C6ST1 did not increase CNS plasticity (Lin et al, 2011). C6ST1 is an enzyme responsible for 6-sulfated CS permissive to axon growth. The significant increase in

its mRNA expression in the injured cord could be a response to the return to a more permissive environment of the CNS (Lin et al, 2011; Miyata et al, 2012) (Fig S3A).

## GlcNAc6ST1 and GlcNAc6ST4 double deficiency reduces the accumulation of monocytes and monocyte-derived macrophages in the scar and penumbra regions of the injured spinal cord

GlcNAc6ST1 and GlcNAc6ST4 are expressed in monocytes and play crucial roles in regulating their migration, activation, and differentiation into macrophages (Tjew et al, 2005). We investigated the potential roles of GlcNAc6ST1 and GlcNAc6ST4 in the regulation of monocyte mobilization to the lesion site after SCI. The monocyte-macrophage markers, CXCR4 and F4/80, were used for validation (Zhang et al, 2011; Werner et al, 2020). Interestingly, the proportions of CXCR4-positive cells and F4/80-positive cells in both scar and penumbra regions 7 d after injury were reduced in GlcNAc6ST1 and GlcNAc6ST4 DKO spinal cords compared with WT, suggesting that monocyte recruitment was reduced in the DKO spinal cords after injury (Fig 3). As monocytes mature into macrophages, they become F4/80 high (Brennan et al, 2022). The F4/80 signal was more pronounced in the WT scars than in the DKO scars (Fig 3). This was probably because of a greater accumulation of macrophages.

## GlcNAc6ST1 and GlcNAc6ST4 double deficiency reduces the abundance of CD68-positive activated macrophages/microglia in the lesion center after injury

We examined whether macrophage/microglia activation is ameliorated in GlcNAc6ST1 and GlcNAc6ST4 DKO mice after injury. CD68 is a marker for the activation of macrophages/microglia in the injured spinal cord. In parallel with CD11b, the mRNA expression of CD68 was up-regulated over 3 mo post-injury (55- and 31-fold increase at 1 and 3 mo post-injury, respectively) in a rodent contusive injury model (Duran et al, 2017) (Fig S4A). Lyzs (lysozyme C-2) is another marker of activated macrophages/microglia. Lyzs mRNA was also up-regulated in the injured cord in a manner similar to CD68 and CD11b over 3 mo post-injury (Fig S4A). Our contusive murine model confirmed the accumulation of Lyzs-expressing cells in the lesion center (Fig S4B). As expected, CD68 immunoreactivity in the lesion center of GlcNAc6ST1/4 DKO mice was less than that of WT (51% of WT) at 3 mo post-injury (Fig 4A). The areas of dorsal white columns, where the corticospinal tracts lie (Watson & Harrison, 2012), and ventral white columns, including the reticulospinal tracts and vestibulospinal tracts, were intensely CD68-immunopositive in the cryo-sections of cords 2-mm rostral to the lesion epicenter at 3 mo post-injury (Fig 4B). The spinal cord sections 2-mm caudal to the lesion center exhibited CD68 immunoreactivity in ventral white columns, with preferential staining in the areas containing the spinothalamic tracts and the descending tracts (Fig 4B). Activated macrophages/microglia situated proximal to the lesion scar at 3 mo post-injury are likely related to the process of retrograde degeneration and dying back of damaged axons. CD68 immunoreactivity in these rostral and caudal positions of the injured GlcNAc6ST1/4 DKO mice was comparable with those in injured WT mice 3 mo post-injury (Fig 4B). The 50% reduction in CD68 staining signals in the lesion center is a position-specific phenotype (Fig 4C). We then asked whether the

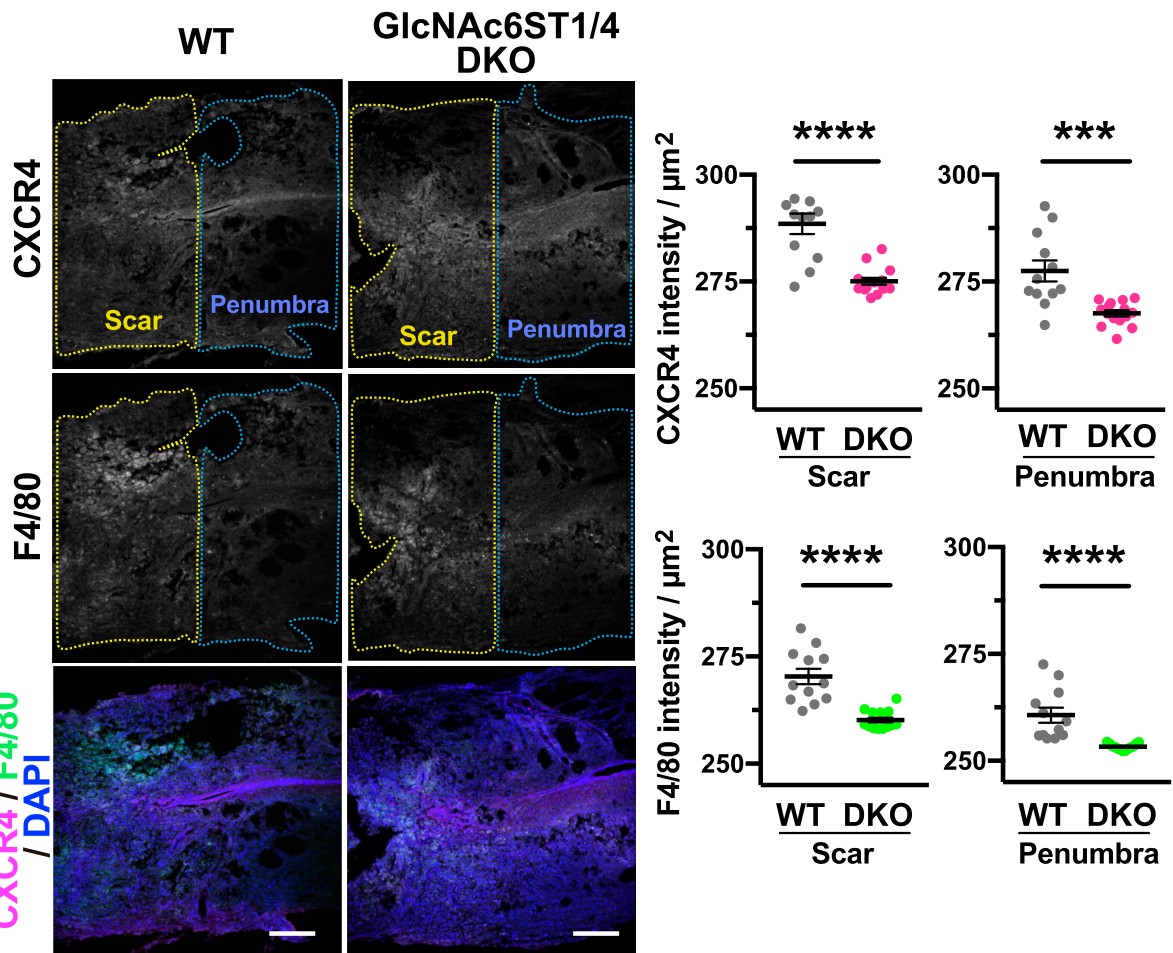

**Figure 3. Reduced accumulation of monocytes and monocyte-derived macrophages in the scar and penumbra regions of injured spinal cord of GlcNAc6ST1 and GlcNAc6ST4 double deficient mice.**
Spinal cord tissues from WT and GlcNAc6ST1 and four doubly deficient (DKO) mice were harvested 7 d after injury. Sagittal frozen spinal cord sections were cut and analyzed by immunohistochemistry. Immunoreactivities for CXCR4 and F4/80, cellular markers for monocytes and monocyte-derived macrophages (Zhang et al, 2011; Werner et al, 2020), are shown in representative images (*n* = 12 sections/3 mice for WT, *n* = 15 sections/3 mice for DKO). *Dashed lines* indicate the lesion scar and penumbra areas of the injured spinal cord. DAPI was used to stain the cell nuclei. Data are means ± s.e.m. ****$P$ < 0.0001, ***$P$ < 0.001. Scale bars: 200 *µ*m. Source data are available for this figure.

double deficiency of GlcNAc6ST1 and GlcNAc6ST4 affects fibrinogenesis of lesion scarring (Dorrier et al, 2021; Sofroniew, 2021) and astrocyte reactivation (Anderson et al, 2016; Tran et al, 2022). Fibronectin, an extracellular component involved in fibrinogenesis, and GFAP, an astrocyte marker, were similarly up-regulated at 3 mo post-injury (2- and 3-fold increase at 1- and 3-mo post-injury, respectively) (Fig S4C). Fibronectin immunoreactivity at the lesion center of GlcNAc6ST1/4 DKO mice was ameliorated to 48% of the WT level, whereas the GFAP level was comparable with those 3 mo post-injury (Fig 4D).

### GlcNAc6ST1 and GlcNAc6ST4 double deficiency facilitates regeneration/sprouting of the 5-HT-positive serotonergic axons after injury

Serotonergic fiber density indicates the regeneration of transected axons and sprouting of spared axon terminals (Tuszynski &

Steward, 2012; Lang et al, 2015). Immunolabeling of 5-hydroxytryptamine (5-HT) provides a method for identifying serotonergic projections (Tuszynski & Steward, 2012; Lang et al, 2015). We wanted to examine if a change in the regeneration/sprouting of serotonergic axons accounts for a part of the improved locomotor recovery seen in GlcNAc6ST1/4 DKO mice after injury. We quantitated the 5-HT immunoreactive fibers at four representative positions of the injured cord, 3 mo post-injury: 5-mm rostral and 2-mm rostral to the lesion epicenter, the lesion center, and 2-mm caudal to the epicenter. In the 5- and 2-mm rostral areas, 5-HT serotonergic axon terminal branches in the grey columns were observed in both WT and DKO mice (Fig 5A). In the lesion center, 5-HT serotonergic fibers were detected in the periphery of the cord, which is the area outside the scar. In the 2-mm caudal area, some 5-HT-positive terminal branches were observed in the ventral horn area of both genotypes (Fig 5A). The levels of 5-HT immunoreactivity at the 5-mm rostral area and the lesion center in DKO mice

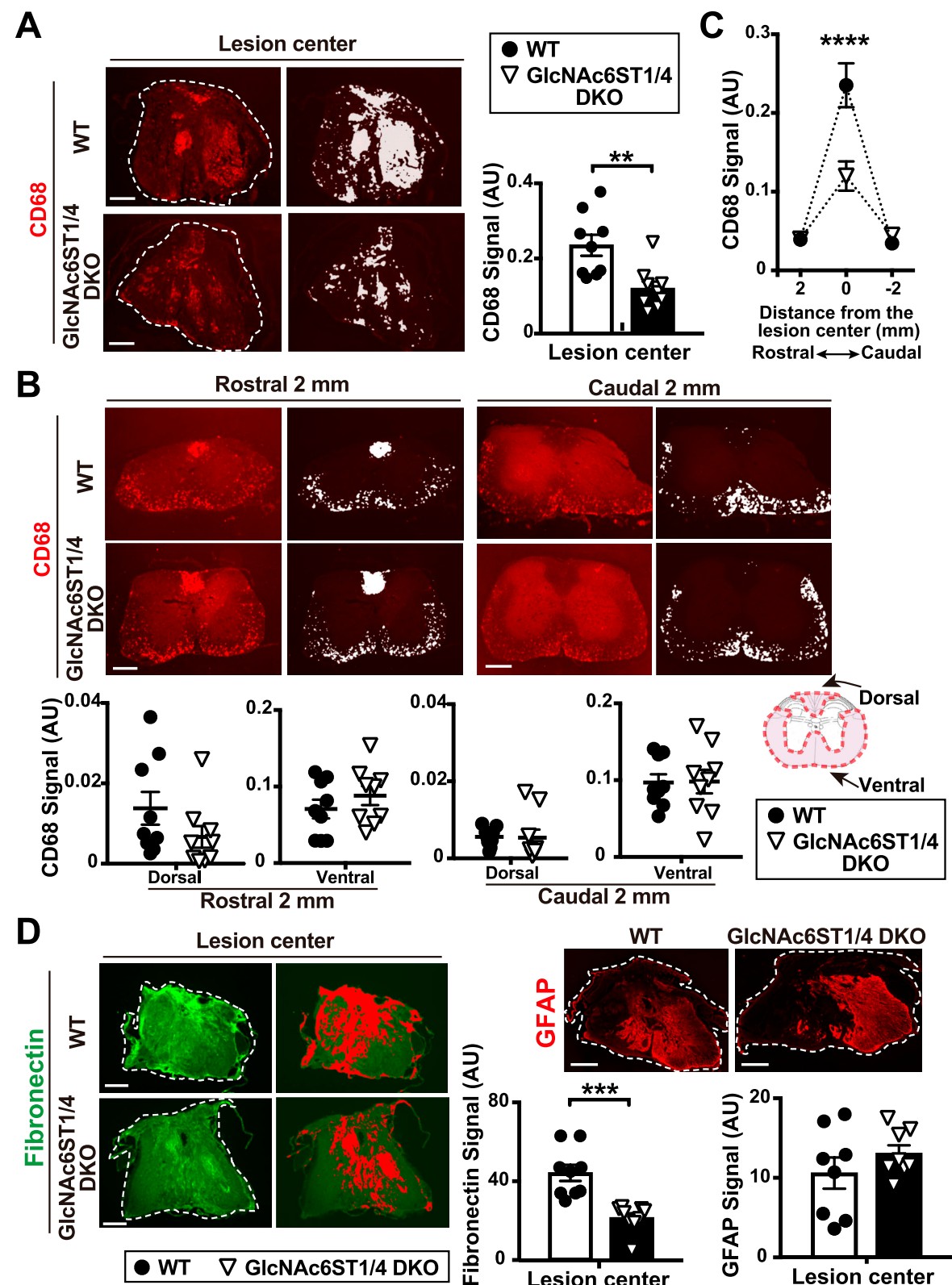

**Figure 4. Reduced activation of macrophages/microglia in the lesion center of injured spinal cord of GlcNAc6ST1 and GlcNAc6ST4 double deficient mice.**
**(A, B)** A contusive spinal cord injury was applied using a 100k-dyn target force at the T10 vertebral level of WT and GlcNAc6ST1 and four doubly deficient (DKO) mice. Immunoreactivity of CD68 (*red*), an activated macrophage/microglia marker, 3 mo post-injury is shown in representative images. **(A, B)** Percentage area of the CD68 signal in traverse sections at the lesion center, and (B) percentage area of the CD68 signal within the dorsal or ventral area 2-mm rostral and 2-mm caudal to the lesion center 3 mo post-injury was measured (*n* = 9 per group). Corresponding micrographs with pseudo-color highlights (*white*) used for quantification are also shown. **(A)** Dashed lines demarcate transverse sections of spinal cords. **(B)** Schematic diagram shows measured areas of the dorsal or ventral white matter. Error bars

were comparable with those in the WT mice (Fig 5B). At the areas 2-mm rostral and 2-mm caudal to the center, the levels of 5-HT immunoreactivity in the DKO were 3.0- and 7.4-fold higher than those in the WT, respectively. The observed difference at the 2-mm caudal site indicated a positive trend ($P$ = 0.08) (Fig 5B). Sparing of white matter correlates with recovery of locomotor function (Noble & Wrathall, 1985). We determined the area of residual white matter stained with Luxol fast blue dyes at the lesion center. The DKO mice had significantly more residual white matter than the WT mice (1.8-fold) 3 mo after injury (Fig 5C).

### GlcNAc6ST1 and GlcNAc6ST4 double deficiency leads to a decrease in the expression of genes encoding type I collagen and ECM components, whereas it concurrently up-regulates gene expression associated with synaptic plasticity within the injured spinal cord

To clarify the transcriptomic changes associated with myeloid lineage cell accumulation and locomotor recovery, we performed whole transcriptome analysis with total RNA-Seq of the injured WT and DKO spinal cords 7 d post-injury. We identified the 93 differentially expressed genes (DEGs) that code for proteins (false discovery rate ≤ 0.05) (Fig 6A and B). The genes of 40 up-regulated and 53 down-regulated were identified in the injured DKO spinal cords (Fig 6A and B). Enrichment analyses of the gene sets by testing the GO terms for the ontology of interest (biological process, molecular function, and cellular component) revealed that the genes involved in the integral component of synaptic membranes were up-regulated in DKO (*Chrna4, Iqsec3, Clstn2, Ank1, Atp2b3, Adcy1, Gabbr1,* and *Atp2b2*) (Fig 6C). The genes involved in cytokine response and cell activation were down-regulated in DKO (*Ccl21b, Lsp1, Cxcl10, Igtp, Sbno2, Igf1, Igtp,* and *Mink1*). Injured DKO also showed the down-regulated genes of type I collagen, collagen fibril organization, and ECM proteins (*Col1a1, Col14a1, Col3a1, Col12a1, Plod2, Loxl1, Lox, Tnn, Eln,* and *Mfap4*) (Fig 6C). Monocytic genes and immune response–related genes of macrophages and microglia (*Ly86, Clec11a,* and *H2-D1*) were down-regulated in DKO, which was somewhat consistent with the result of reduced levels of monocytes and monocyte-derived macrophages described above. To gain more insight into which type of glycan synthesis is impaired in the injured GlcNAc6ST1/4 DKO spinal cords, we performed glycomic analysis with GALAXY (Yagi et al, 2005, 2022). It was found that GlcNAc6ST1/4 double deficiency resulted in the loss of Galß1-4GlcNAc(6S)ß1-branch and Galß1-4GlcNAc(6S)(α1-3Fuc)ß1-branch containing *N*-glycans but the preservation of Gal(6S)ß1-4GlcNAc(α1-3Fuc)ß1-branch containing *N*-glycans in the injured spinal cords (Fig 6D).

## Discussion

The GlcNAc6ST family in mice consists of GlcNAc6ST1, GlcNAc6ST2, GlcNAc6ST3, and GlcNAc6ST4 (Uchimura & Rosen, 2006). With the exception of GlcNAc6ST2 (encoded by the *Chst4*), the other three sulfotransferases are expressed in the injured spinal cord. In this study, we demonstrated that a single gene deficiency in any of these spinal cord-expressing sulfotransferases results in locomotor recovery comparable with that in WT mice after injury. When mice were doubly deficient in GlcNAc6ST1 and GlcNAc6ST4, both of which are expressed in CD11b-positive myeloid lineage cells, significant improvements in locomotor recovery were observed. This enhancement was accompanied by a reduction in the densities of infiltrating monocytes and macrophages within both the scar and penumbra regions. In addition, there was a decrease in the activation of macrophages/microglia in the lesion center, an increase in serotonergic axon density rostral and caudal to the lesion, and a greater preservation of white matter in the lesion center.

GlcNAc6ST1 and GlcNAc6ST4 are expressed in blood monocytes and are involved in the invasion of these cells into injured tissues (Maiti et al, 1998; Tjew et al, 2005). The GlcNAc6ST4 gene-expressing cells that accumulated in close proximity with the blood vessels in the lesion center, as revealed by X-gal staining, most likely include such myeloid cells. The scRNA-seq data (Hamel et al, 2023) indicated that cells expressing both the GlcNAc6ST1 and GlcNAc6ST4 genes include infiltrating monocyte-derived macrophages in the injured spinal cord. We found that the accumulation of monocytes/macrophages in the lesion site was significantly reduced in the DKO spinal cords after injury. It is strongly suggested that molecules modified with GlcNAc6ST1/4-dependent sulfated glycans control the mobilization and activity of these myeloid lineage cells. Statistically significant reductions detected in DKO differed by less than 10%. This is very interesting and may reflect the finding that only a small subpopulation was affected. Secondary damage because of SCI includes retractive effects on axons by blood monocyte-derived macrophages (Evans et al, 2014). These detrimental effects could be attenuated by the reduced accumulation of CD68-positive cells at the lesion center of the injured GlcNAc6ST1/4 DKO spinal cord. The basis for the comparable immunoreactivity of the anti-CD68 antibody at the rostral and caudal sites of the lesion center in DKO mice remains to be elucidated. The expression level of fibronectin, a fibroblastic scar component, was significantly reduced in the DKO lesion center, whereas astrocyte reactivation was comparable with that of WT mice. Fibrotic scarring occurs in response to circulating inflammatory cell infiltration in the CNS (Dorrier et al, 2021; Sofroniew, 2021). The

show s.e.m. **$P$ < 0.01. **(C)** Ordinary two-way ANOVA with distance and genotype as factors revealed significant effects on the activation of macrophages/microglia 3 mo post-injury (distance: $F_{2, 48}$ = 58.58, $P$ < 0.0001; genotype: $F_{1, 48}$ = 8.38, $P$ < 0.01; interaction between distance and genotype: $F_{2, 48}$ = 11.26, $P$ < 0.0001). Post hoc Bonferroni's test showed a significant reduction in CD68 immunoreactivity in the lesion center of GlcNAc6ST1/4 DKO mice 3 mo post-injury. Data are means ± s.e.m. ****$P$ < 0.0001. **(D)** Immunoreactivity of fibronectin (*green*), a component of fibroblastic scarring, and glial fibrillary acidic protein (GFAP) (*red*), a marker of reactive astrocytes, in the lesion center 3 mo post-injury are shown. Corresponding micrographs with pseudo-color highlights (*red*) are also shown for fibronectin. Dashed lines demarcate the sections of spinal cord. Percentage area of the stained signal was determined in transverse sections at the lesion center 3 mo post-injury (fibronectin: $n$ = 9 per group; GFAP: $n$ = 8 per group). Data are means ± s.e.m. ***$P$ < 0.001. Scale bars: 250 $\mu$m.
Source data are available for this figure.

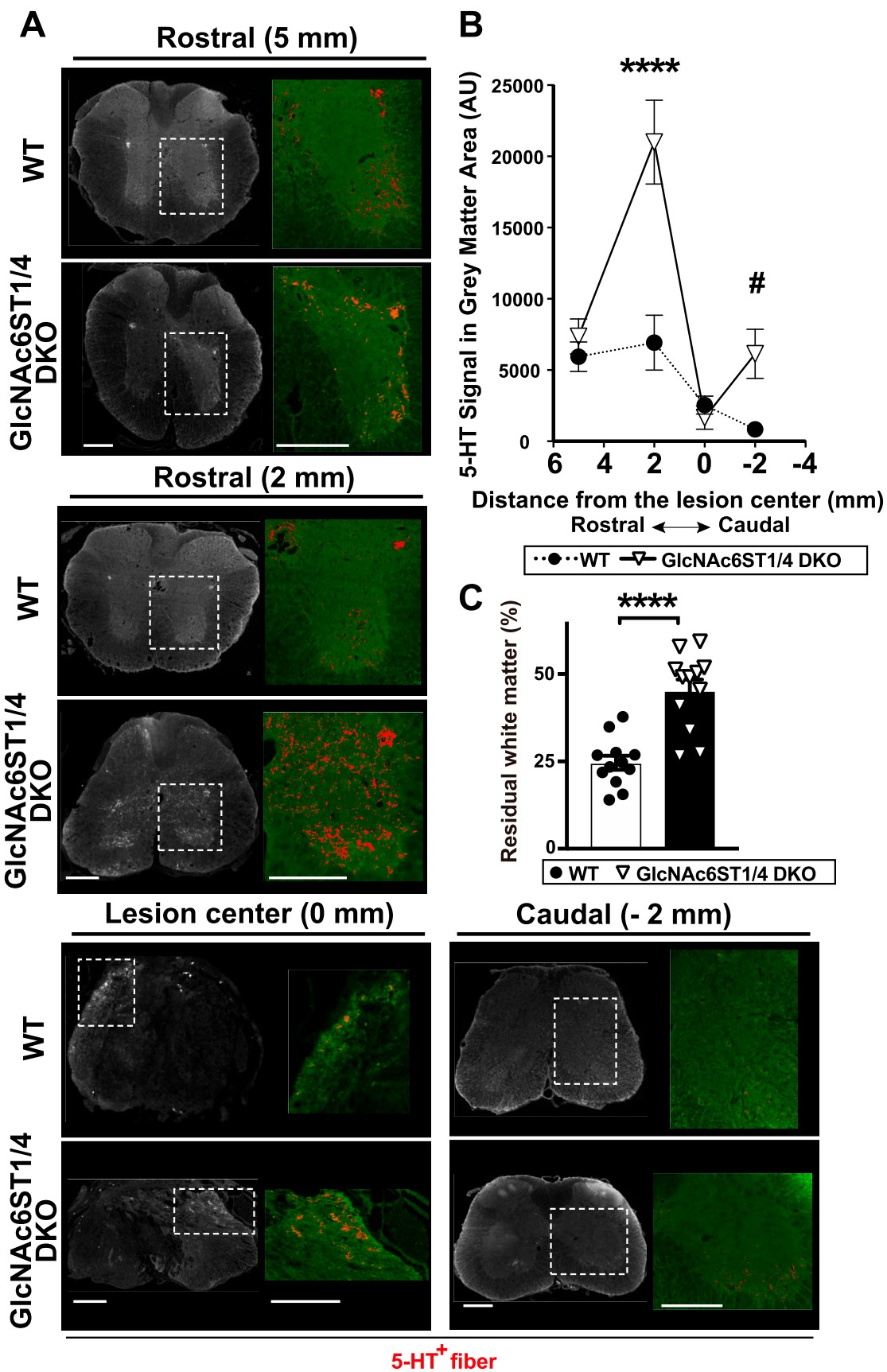

5-HT⁺ fiber

decreased expression of fibronectin in DKO can be attributed to reduced monocyte infiltration. Additional validation of immunohistochemistry conducted on longitudinal sections of the injured spinal cord may serve to corroborate our findings. As shown in this study, GlcNAc6ST1 and GlcNAc6ST4 are also expressed in CD11b-negative cell fractions in the injured spinal cord. It is unclear whether both GlcNAc6ST1 and GlcNAc6ST4 are also cell-autonomous regulators of astrocyte/fibrotic scar formation in non-myeloid cells. Expression of these enzymes in fibroblasts, pericytes (Goritz et al, 2011; Dias et al, 2018), and reactivated astrocytes (Anderson et al, 2016; Hara et al, 2017), as well as their contributions to cell functions in the lesion scarring would be another important area for future study. Cell-type–specific conditional knockout techniques would be useful to test these possibilities.

Sprouting of the serotonergic axonal tracts correlates with motor functional recovery after SCI (Rossignol & Dubuc, 1994; Tuszynski & Steward, 2012; Ghosh & Pearse, 2014; Lang et al, 2015). Our findings indicate a significant increase in the number of 5-HT-positive serotonergic fibers in the ventral horn, both 2-mm rostral and 2-mm caudal to the lesion center, in the injured GlcNAc6ST1/4 DKO mice, with a favourable trend observed in the latter region. This is pertinent to serotonergic neural plasticity. The local sprouting may promote functional recovery through propriospinal interneurons, which can form relay connections that bypass the lesion site (Bareyre et al, 2004; Vavrek et al, 2006). However, further investigation is needed to address whether the increase in serotonergic fibers, rostral and caudal to the injury, contribute to the improvement of locomotor recovery observed in the DKO mice. Additional validation may support the presence of 5-HT signals in the peripheral areas of the lesion centers of WT and DKO specimens. Our results support the idea that reduced accumulation of infiltrated monocytes and CD68-positive activated macrophages/microglia resulted in reduced secondary axonal dieback of serotonergic neural fibers in the DKO mice. Thus, double deficiency of GlcNAc6ST1 and GlcNAc6ST4 has implications for serotonergic neural plasticity after SCI. In injured DKO mice, greater white matter sparing in the chronic stage of injury was demonstrated by Luxol fast blue staining of myelinated axons. Better preserved white matter axons may also contribute to the recovery of locomotor function in the injured DKO mice.

Transcriptome analysis with total RNA-Seq revealed that the down-regulation of the expression of monocytic genes and immune response-related genes of macrophages and microglia in DKO was well consistent with the finding that the levels of recruited monocytes and monocyte-derived macrophages were significantly reduced in the injured DKO spinal cords 7 d post-injury. GO term

enrichment analyses revealed that gene clusters related to the integral component of the synaptic membrane were up-regulated in the injured DKO spinal cords. It is conceivable that synaptic plasticity is enhanced and synapses are strengthened in the injured DKO spinal cord, which may also be effective in restoring motor function. The amount of ECM constituent molecules, including type I collagen, and the structure of the ECM are regulatory factors in neuronal and synaptic plasticity (Li et al, 2020; Nguyen et al, 2020; Wahane et al, 2021; Bi et al, 2024 *Preprint*). ECM negatively regulates the plasticity after injury by interacting with reactive astrocytes (Hara et al, 2017). The total RNA-Seq data showed that the expression levels of genes encoding type I collagen and ECM-related molecules were down-regulated in the injured DKO spinal cords. This finding supports the possibility that improved locomotor function is associated with reduced ECM synthesis. The reduced expression of ECM protein genes in DKO may be partially explained by the decrease in infiltrating monocytes into the CNS lesion (Dorrier et al, 2021; Sofroniew, 2021). Glycomic analysis showed that all 6-sulfated GlcNAc-containing *N*-glycans detectable in the injured spinal cords were abolished in DKO, whereas 6-sulfated Gal-containing *N*-glycans were retained. It is conceivable that these 6-sulfated GlcNAc-containing *N*-glycans are present in the cell surface of monocytes and that they are involved in monocyte recruitment. One candidate molecule is CD44. Certain Siglecs are known to bind to 6-sulfated GlcNAc-containing glycans (Duan & Paulson, 2020). Therefore, it is possible that there is an altered function of Siglec in *cis* expressed on CD11b-positive myeloid cells leading to the reduced level of microphage/microglia activation in the DKO SCI. The possible contribution of the non-reducing end structures of the GlcNAc-6-sulfated glycans, such as sialic acid, would be another topic of research on SCI (Mountney et al, 2010, 2013).

GlcNAc6STs mediate C-6-sulfation of GlcNAc residues within *N*- and *O*-linked glycans of glycoproteins and keratan sulfate polysaccharide, a class of sulfated glycosaminoglycans. GlcNAc6ST1 is involved in blood monocyte functions by sulfation modification of the CD44 cell-surface glycoprotein as described above, in addition to regulating lymphocyte homing to lymph nodes by synthesizing 6-sulfo sialyl Lewis X on sialomucin *O*-glycans in concert with GlcNAc6ST2 (Uchimura et al, 2005). In the CNS, GlcNAc6ST1 has been shown to be implicated in the synthesis of phosphacan/Ptprz1 keratan sulfate present in the extracellular space of the developing brain but not in the adult (Hoshino et al, 2014; Takeda-Uchimura et al, 2015). Under pathological conditions, we found that a short Gal/GlcNAc-disulfated oligosaccharide recognized by the 5D4 antibody (Zhang et al, 2017) is extensively induced in mouse microglia. This epitope expression is completely eliminated upon

**Figure 5. Increased level of 5-HT+ serotonergic neuron density in the injured spinal cord of GlcNAc6ST1 and GlcNAc6ST4 double deficient mice.**
**(A, B)** A contusive spinal cord injury using a 100k-dyn target force at the T10 vertebral level was applied to WT, and GlcNAc6ST1 and four doubly deficient (DKO) mice. **(A)** Immunoreactivity of 5-HT at the indicated position rostral or caudal to the lesion center 3 mo post-injury is shown in representative images. Inset area in each image is magnified with pseudo-color signals of 5-HT (*red*). **(B)** The 5-HT signal area within the grey matter area of transverse sections 3 mo post-injury was measured. Ordinary two-way ANOVA with distance and genotype as factors revealed significant effects on increase in 5-HT serotonergic axon density (distance: $F_{3, 32}$ = 23.55, $P < 0.0001$; genotype: $F_{1, 32}$ = 20.81, $P < 0.0001$; interaction between distance and genotype: $F_{3, 32}$ = 9.088, $P < 0.001$). Post hoc Bonferroni's test showed a significant increase in the 5-HT immunoreactivity in the area 2-mm rostral to the lesion center of GlcNAc6ST1/4 DKO mice 3 mo post-injury. The difference at 2-mm caudal to the lesion center showed a statistical trend ($^{\#}P = 0.08$) (n = 5 per group). **(C)** Residual white matter was measured using Luxol fast blue staining on transverse spinal cord sections from the lesion center of WT and DKO 3 mo after injury. DKO has more spared white matter (n = 12 sections/3 mice per group). Data are means ± s.e.m. ****$P < 0.0001$. Scale bars: 250 $\mu$m. Source data are available for this figure.

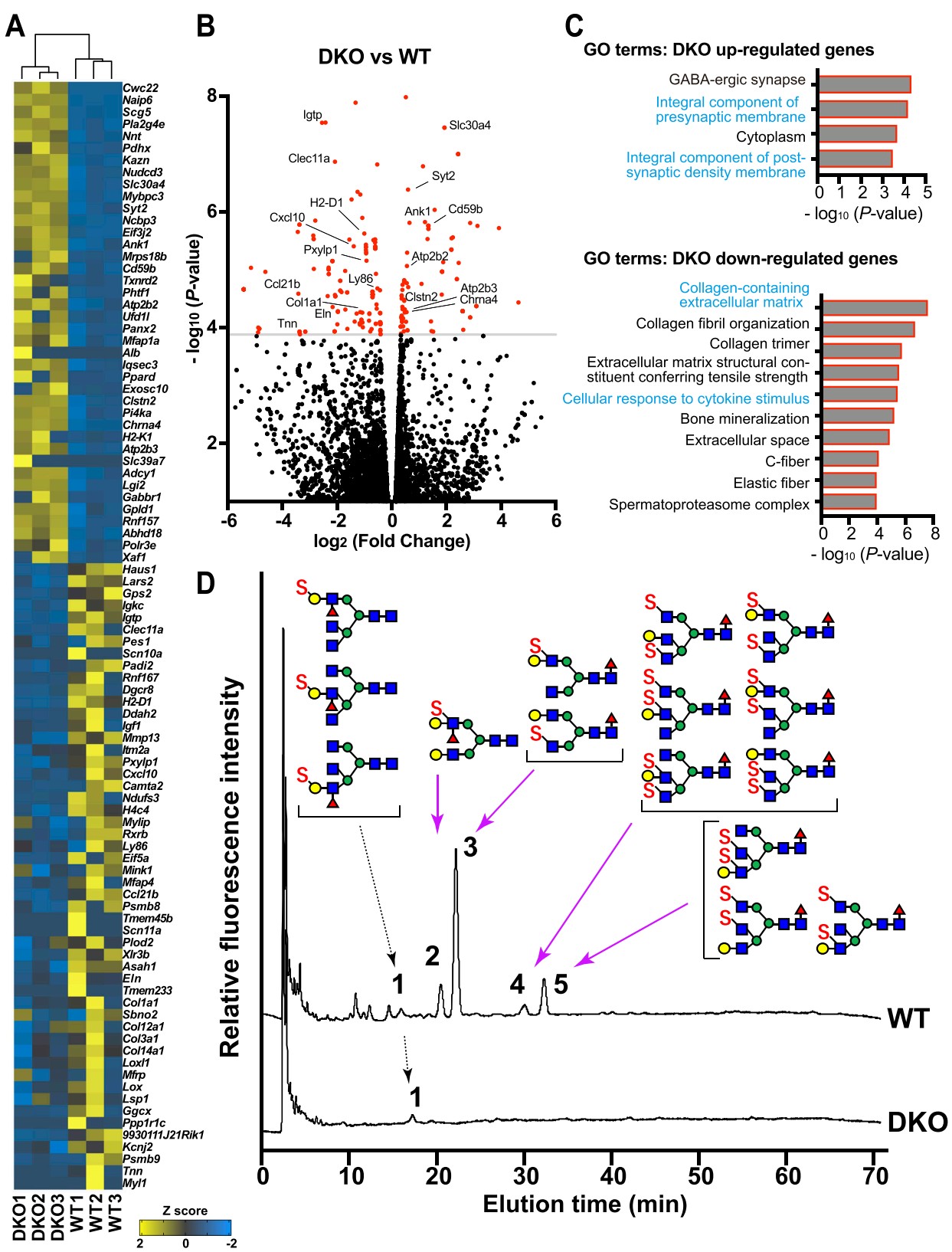

**Figure 6. Effects of GlcNAc6ST1 and GlcNAc6ST4 double deficiency on the transcriptome and glycome in the injured spinal cord.**
**(A)** Heatmap of the 93 DEGs (false discovery rate q < 0.05, ranked by P-value) in the injured GlcNAc6ST1/4 DKO spinal cords 7 d post-injury (n = 3 per group). **(B)** Volcano plot based on the log₂ fold change showing genes in the injured DKO spinal cords 7 d post-injury. The grey line represents the q < 0.05 cutoff. **(C)** Enriched gene ontology (GO) terms of up- and down-regulated DEGs in the injured DKO spinal cords. **(D)** GALAXY glycomic analysis (Yagi et al, 2005, 2022) showing high-performance liquid chromatography elution profiles of pyridyl-2-aminated-N-glycans derived from the injured spinal cords of WT and GlcNAc6ST1/4 DKO mice (7 d post-injury) on an

GlcNAc6ST1 deletion (Zhang et al, 2017). Similar to our previous observation, the 5D4 epitope is elicited in macrophages and microglia in the spinal cord after injury (Jones & Tuszynski, 2002). In the present study, we initially expected that the elimination of the 5D4-reactive short glycan of myeloid cells by single deficiency of GlcNAc6ST1 would show amelioration of locomotor recovery after SCI. However, the outcomes did not align with the anticipated results. GlcNAc6ST4 shows expression profiles and substrate specificity different from the other three members of the family (Uchimura et al, 2000, 2002a). It is unlikely that GlcNAc6ST4 mediates GlcNAc-6-sulfation of keratan sulfate polysaccharide (Narentuya et al, 2019). Because the double deficiency of GlcNAc6ST1 and GlcNAc6ST4 showed better locomotor improvement than either single deficiency, it is plausible that the GlcNAc6ST1/4-mediated GlcNAc-6-sulfated oligosaccharides of the myeloid cell glycoproteins have a functional involvement dominant to that of the 5D4-reactive short glycan. These GlcNAc6ST1/4-mediated sulfated oligosaccharides, distinct from keratan sulfate polysaccharide, include the GlcNAc-6-sulfated *N*-glycans that were eliminated in the injured DKO spinal cord as revealed by the GALAXY glycomic analysis. The depletion of both the 5D4-reactive short glycan and the GlcNAc-6-sulfated *N*-glycans may reflect the better locomotor recovery in the DKO of GlcNAc6ST1 and GlcNAc6ST4 than in either single KO. Keratan sulfate polysaccharide was previously reported to restrict axonal regeneration (Ito et al, 2010). The fact that GlcNAc6ST3, but not GlcNAc6ST1, is a major enzyme of keratan sulfate polysaccharides in the adult CNS (Takeda-Uchimura et al, 2015; Narentuya et al, 2019) motivated us to examine if deletion of GlcNAc6ST3 would promote neuronal plasticity in motor integration and facilitate locomotor recovery after SCI. Unexpectedly, the GlcNAc6ST3 deficiency did not improve locomotor function after contusive SCI, and keratan sulfate polysaccharides were down-regulated in the injured spinal cord (Fig S5), supporting the idea that GlcNAc-6-sulfation of keratan sulfate polysaccharide is not a limiting factor for motor integration after contusive injury. Mice deficient in Gal-6-*O*-sulfotransferase KSGal6ST or C6ST1 were also examined to assess the possible contribution of Gal/GlcNAc-disulfated keratan sulfates to improvement of motor recovery after SCI. Deficiency of either of these Gal-6-*O*-sulfotransferases did not result in improved locomotor recovery after SCI, suggesting that Gal-6-sulfation modification of keratan sulfate polysaccharide does not serve as a barrier to axonal regrowth and sprouting in vivo. Whether keratan sulfate polysaccharide restricts neuronal regrowth is still up for debate, as results have been conflicting (Ito et al, 2010). In addition to our findings, recent studies have shown that keratan sulfate polysaccharide has no restrictive effects on neuronal regrowth (Hering et al, 2020) and is identified as an axon regeneration promoting molecule that is up-regulated in the pro-regenerative environment of the SCI lesion in the spiny mice (*Acomys*) (Seifert et al, 2012; Nogueira-Rodrigues et al, 2022).

Keratan sulfate polysaccharide present in the neuropil is more likely to be a promoting rather than an inhibitory molecule. B3Gnt7, an enzyme for keratan sulfate polymerization, is identified as an enhancer of axon regeneration in vivo (Nogueira-Rodrigues et al, 2022). Rather than the degree of sulfation, the length of keratan sulfate may be a factor that promotes axon regeneration; testing this issue in B3Gnt7-KO (Takeda-Uchimura et al, 2022) is a future perspective. We have generated mice that are triple deficient in GlcNAc6ST1, GlcNAc6ST3, and GlcNAc6ST4 (Takeda-Uchimura et al, 2024). Studying these mice for inflammatory responses and locomotor recovery after SCI would provide additional details on these issues.

Taken together, GlcNAc6ST1/4-dependent 6-sulfated GlcNAc glycans expressed on the cell surface of recruited monocytes, macrophages, and microglia are likely to control their cellular functions, leading to the exertion of the effects on neural plasticity. A potential future therapeutic strategy after SCI is the administration of inhibitors of the GlcNAc6ST1 and GlcNAc6ST4 enzymes. The development of specific inhibitors of these enzymes is also a future challenge.

# Materials and Methods

### Animals

C57BL/6J mice were purchased from SLC Inc. GlcNAc6ST1-deficient (KO) ($Chst2^{-/-}$) mice (Uchimura et al, 2004), GlcNAc6ST4-KO ($Chst7^{-/-}$) mice (Narentuya et al, 2019), and GlcNAc6ST3-KO ($Chst5^{-/-}$) mice (Hayashida et al, 2006) were generated as described previously. KSGal6ST-KO ($Chst1^{-/-}$) mice (Patnode et al, 2013) and C6ST1-KO ($Chst3^{-/-}$) mice (Uchimura et al, 2002b) were obtained as described previously. Mice doubly deficient in GlcNAc6ST1 and GlcNAc6ST4 were generated by breeding GlcNAc6ST1 and GlcNAc6ST4 double heterozygous mice. All KO mice were maintained on a C57BL/6J genetic background. Males and females of all genotypes (3.5–6 mo of age) were used for experiments. All mice were maintained under controlled SPF environmental conditions and provided with standard nourishment and water in the animal facilities of Nagoya University Graduate School of Medicine and Research Institute of Environmental Medicine. All experiments were approved by the Animal Research Committee of Nagoya University and performed in accordance with the guidelines of Nagoya University.

### Surgical procedures

Mice were anesthetized with pentobarbital sodium (50 mg/kg, i.p.). A laminectomy was performed at the T10 vertebral level. The dura mater was exposed. A commercially available SCI device (Infinite Horizon Impactor; Precision Systems and Instrumentation, LLC,

---

octadecyl silica column of the asialo-anionic fractions separated by a diethylaminoethyl column. The *dashed arrow* indicates the peak that was detected in both WT and DKO spinal cords (peak 1). The *pink arrows* indicate the positions of the peaks that were not detected in the DKO spinal cord profile (peaks 2–5). The glycan structures obtained by the GALAXY analysis are presented. Representative data are shown (*n* = 2 per group). Note that all GlycNAc-6-sulfated glycans except Gal-6-sulfated glycans detected in the injured WT spinal cords were eliminated in DKO. Symbols denote the following: C-6 sulfate (S), *N*-acetylglucosamine (GlcNAc; *blue square*), galactose (Gal; *yellow circle*), mannose (*green circle*), and fucose (*red triangle*).
Source data are available for this figure.

Fairfax Station) was used to produce a contusive SCI, using a 100k-dyn force. Post-anesthesia monitoring was carried out, whereas the animal was on a surgical warming pad and until fully ambulatory. Postoperative care included manual expression of the bladder once daily until 2 wk post-injury. Oral administration of the prophylactic antibiotic treatment (1:500 diluted Bactramin in acidified water; Chugai Pharmaceutical) was carried out for 1 wk post-injury to avoid bladder infections and surgical site infections. Postoperative monitoring was conducted daily and recorded for each animal. The genotype of each animal was blinded during all surgical processes and postoperative monitoring (M.M., T.O., A.M.).

### Preparation of the spinal cord for histology and biochemical assays

Injured and sham-operated (laminectomy only) mice were anesthetized and then transcardially perfused with PBS. For biochemical and gene expression analyses, each spinal cord was removed from the vertebral column, cut into segments on ice, snap-frozen with liquid nitrogen, and then stored at −80°C. For immunohistochemical analyses, the anesthetized mice were perfused with PBS followed by PBS containing 4% PFA. Each vertebral column was removed and post-fixed overnight in 4% PFA/PBS at 4°C. The spinal cord was then extracted from the vertebral column and post-fixed overnight. The cord was equilibrated in 30% sucrose in PBS, cut into 1-mm blocks rostral or caudal to the middle of the lesion center, and then embedded into Tissue-Tek (O.C.T. compounds; Sakura).

### Isolation of CD11b-positive primary cells

CD11b-positive primary cells were isolated from injured and sham-operated mice 7 d post-operation. Mice were perfused transcardially with ice-cold PBS after heavy anesthetization. A 9-mm length of spinal cord centered over the epicenter, or a matching segment of the uninjured cord, was dissected out. The spinal cord was fragmented using a scalpel and then incubated in a tube containing a solution with 5 mg/ml collagenase (Wako) and 10 $\mu$g/$\mu$l DNase I (Roche Diagnostics) for 45 min at 37°C. The tube was inverted every 10 min. The solution was gently pipetted to disrupt the tissue. The reaction mixture was filtered through a 70-$\mu$m cell strainer to remove visible debris, and the flow-through fraction was centrifuged at 190$g$ for 10 min. The cell pellets were resuspended in 2 ml of PBS with 0.5% BSA and 2 mM EDTA. The cell mixture was then incubated with CD11b magnetic beads (Miltenyi Biotec) for 15 min at 4°C. CD11b bead-bound cells were collected by centrifugation as described above. The collected CD11b-positive cells were snap-frozen and stored at −80°C.

### Quantitative real-time PCR

The total RNA was extracted from frozen mouse spinal cord tissues and primary CD11b-positive cells using the TRIzol Reagent (Thermo Fisher Scientific), according to the manufacturer's instructions. DNase I-treated RNA (5 $\mu$g) was subjected to reverse transcription in 100 $\mu$l of reaction buffer with RNase inhibitor (Roche Diagnostics), random primers, SuperScript II Reverse Transcriptase (Thermo Fisher Scientific), and a mixture of dNTPs (Roche Diagnostics). Quantitative real-time PCR was performed with SYBR qPCR Mix (Toyobo) and the Mx3000P Real-Time QPCR System (in the Materials and Methods section), quantified using qRT-PCR (in Fig 1). The following procedure was used: 1 cycle of 95°C for 10 s, and 40 cycles of 95°C for 5 s and 60°C for 30 s. The relative expression levels for each mRNA were normalized to the level of *Gapdh* mRNA, calculated using the $2^{-\Delta\Delta Ct}$ method. An appropriate control was set as the calibrator. Triplicate real-time PCR reactions were used for analysis. The primers for GlcNAc6ST1 (*Chst2*), GlcNAc6ST2 (*Chst4*), GlcNAc6ST3 (*Chst5*), GlcNAc6ST4 (*Chst7*), KSGal6ST (*Chst1*), and C6ST1 (*Chst3*) were used as described previously (Foyez et al, 2015; Zhang et al, 2017). The primer sequences were as follows: 5′-AGGAG-CACCTCGGTATCAGCA-3′ (forward) and 5′-CCATCAGCGTCCATGTC-CAC-3′ (reverse) for CD11b (*Itgam*), 5′-ACCAGCTTACGGCCAACAGTG-3′ (forward) and 5′-TGTCTATACGCAGCCAGGTTGTTC-3′ (reverse) for glial fibrillary acidic protein (*Gfap*), and 5′-TCTTGTGCAGTGCCAGCCTCGT-3′ (forward) and 5′-TCACAAGAGAAGGCAGCCCTGG-3′ (reverse) for glyceraldehyde 3-phosphate dehydrogenase (*Gapdh*).

### Immunohistochemistry

Frozen spinal cord tissues were cut into 16-$\mu$m–thick transverse sections on a cryostat and collected on MAS-coated glass slides (SF17293; Matsunami). The sections were stored at −80°C for additional processing. At least three sections per animal at each distance, rostral or caudal to the lesion center, were stained and analyzed. Sections were air-dried for 30 min, rinsed with PBS to remove O.C.T., and then blocked in 3% BSA and 0.1% Triton X-100 in PBS for 15 min at RT. Sections were then incubated overnight at 4°C with primary antibodies: rat anti-mouse CD68 (1:100 dilution, #MCA1957; Bio-Rad Laboratories), rabbit anti-fibronectin (1:100 dilution, #F3648; Sigma-Aldrich), or Cy3-conjugated mouse anti-GFAP (1:250 dilution, #C9257; Sigma-Aldrich). Sections were washed with PBS and then incubated at RT for 30 min with secondary antibodies: Cy3-conjugated goat anti-rat IgG (1:500 dilution; Jackson ImmunoResearch Laboratory) for CD68, or AlexaFluor 488-conjugated goat anti-rabbit IgG (1:500 dilution; Jackson ImmunoResearch Laboratory) for fibronectin. After washing with PBS, the stained sections were mounted in FluorSave Reagent (Merck). Signals were visualized and captured using a fluorescent microscope (model BZ-9000; Keyence) at the same exposure setting for each antibody. The exposure time was set to 1/40 s (CD68) or 1/200 s (GFAP) with the low photobleach setting and standard resolution sensitivity. The baseline fluorescent intensity observed in the grey matter was set as the threshold, and staining signals that exceeded this threshold were subsequently quantified using the BZ-II analysis application software (Keyence).

### X-gal staining

To determine the lacZ gene-expressing cells, cryo-cut spinal cord sections were fixed and stained with the reagents of a

ß-galactosidase reporter gene staining kit (Sigma-Aldrich), according to the manufacturer's instructions.

## Analysis of monocytes/monocyte-derived macrophages

Sagittal frozen spinal cord sections of 16-$\mu$m thickness were cut on a cryostat microtome and collected on MAS-GP type A-coated slides (Matsunami). For heat-induced epitope retrieval, spinal cord sections were rinsed three times with PBS and then incubated in citric acid buffer (1.8 mM trisodium citrate dihydrate, 8.9 mM citric acid monohydrate) with microwave heating for 8 min. The sections in the buffer were allowed to cool on ice for 30 min. Sections were washed three times with PBS and then incubated in a blocking solution containing 10% (vol/vol) normal goat serum (Vector Laboratories) and 0.05% Triton X-100 for 1 h at RT. Sections were incubated overnight at 4°C with 1:500 rabbit anti-CXCR4 (#GTX22074; GeneTex) in PBS containing 5% (vol/vol) normal goat serum and 0.025% (vol/vol) Triton X-100. Sections were washed with PBS and then incubated with 1:250 PE-Cyanine5-conjugated rat anti-F4/80 (Clone BM8; #15-4801-82; Invitrogen) and 1:500 Alexa Fluor Plus 594-labeled goat anti-rabbit IgG (highly cross-absorbed; #A32740; Invitrogen) in PBS containing 5% (vol/vol) normal goat serum and 0.025% (vol/vol) Triton X-100 for 2 h at RT. After washing three times with PBS, the sections were incubated with 1:1,000 DAPI (Dojindo Laboratories) in PBS for 20 min at RT. The sections were embedded in ProLong Diamond Antifade Mountant (Invitrogen). Fluorescence images were acquired using a confocal super-resolution microscope (SpinSR10, Olympus) equipped with a confocal spinning disk confocal laser scanning unit (CSU-W1 SoRa, Yokogawa), an UPlanXApo 10 X objective lens (Olympus) and a sCMOS camera (ORCA-Flash4.0, Hamamatsu Photonics KK). Scanning parameters were unified across specimens. Images were analyzed using cellSens imaging software (Olympus) to quantify fluorescent signals in the scar and penumbra regions of injured spinal cord. Four to five sections at the scar and penumbra sites of each mouse were used for analysis.

## Analysis of 5-HT-positive serotonergic fiber tracts

Segments of frozen spinal cord were cut into 16-$\mu$m–thick transverse sections on a cryostat. At least three sample sections per animal at each distance rostral or caudal to the lesion center were stained and analyzed. The sections were air-dried for 30 min, rinsed with PBS, and then incubated in 5% normal donkey serum in PBS containing 0.1% Triton X-100 for 2 h at RT. They were then incubated with a goat anti-5-hydroxytryptamine (5-HT; serotonin) antibody (1:500 dilution, #20079; ImmunoStar) for 36 h at 4°C. Afterwards, the sections were extensively washed with PBS and then incubated with AlexaFluor 488-conjugated donkey anti-goat IgG (1:250 dilution; Invitrogen) for 3 h at RT. Finally, the sections were washed extensively with PBS and mounted in FluorSave Reagent. Stained signals of 5-HT+ serotonergic fibers were visualized and captured using a fluorescent microscope (model BZ-9000; Keyence). During the image acquisition process, we captured 40× split images, which were subsequently consolidated. The resulting consolidated images were analyzed. The exposure time was set to 1/20 s with the low photobleach setting and standard resolution sensitivity. Fluorescent 5-HT signals within the grey matter of cross sections were quantified using the BZ-II software (Keyence) with the same values of threshold and correction, and the measurement target range setting.

## Luxol fast blue staining

Frozen transverse spinal cord sections of 16-$\mu$m thickness were cut on a cryostat and stained with 0.1% Luxol fast blue (Sigma-Aldrich) in 0.5% acetic acid at 60°C for 3 h. The sections were then immersed in 0.05% lithium carbonate for 3 min to visualize the white matter. The sections were microscopically examined in the bright field of a fluorescence microscope and analyzed (Model BZ-X710; Keyence). Luxol fast blue staining area was quantified using the BZ-X software (Keyence). Four sections at the lesion center of each mouse were used for analysis.

## Locomotor Recovery based upon the BMS over 8 wk post-injury

Locomotor outcomes based on coordination, trunk stability, and paw position were rated in a blinded manner. The evaluator (A.M.) and surgeons (M.M., T.O.) were unaware of the genotypes of the subjected mice. Spinal cord injured mice that showed more than 1 point-scale difference in the BMS score (Basso et al, 2006) at day 1 post-injury were considered to have sustained milder injuries and were excluded from the entire study.

## RNA-Seq transcriptome analysis

Spinal cord tissues were harvested at 8-mm length centered on the injury site 7 d after injury. Total RNA was extracted from the tissues using the RNeasy Lipid Tissue Mini Kit (QIAGEN). Total RNA quality was analyzed using a 5200 Fragment Analyzer System and Agilent HS RNA Kit (Agilent Technologies), and libraries for RNA sequencing were prepared using MGIEasy RNA Directional Library Prep Set (MGI Tech). Quantities and qualities of the libraries were determined by using a Qubit 3.0 Fluorometer and dsDNA HS Assay Kit (Thermo Fisher Scientific), and a Fragment Analyzer and dsDNA 915 Reagent Kit (Agilent Technologies). The dsDNAs in the libraries were then converted into single-stranded circular DNA libraries using the MGIEasy Circularization Kit (MGI Tech). DNA nanoballs were prepared by using DNBSEQ G 400 RS High-throughput Sequencing Kit (MGI Tech) and sequenced by using a DNBSEQ G400 sequencer (MGI Tech) with 2 × 150 bp paired-end reads. Adaptor sequences and low-quality sequences were removed using Cutadapt v1.9.1 (https://cutadapt.readthedocs.io) and sickle v1.33. Reads were then aligned to the *Mus musculus* GRCm38.p6 genome using Hisat2 v2.2.0 (https://daehwankimlab.github.io/hisat2/), summarized with featureCounts v2.0.0 (https://subread.sourceforge.net/featureCounts.html), and converted to reads per kilobase million with normalization using Trimmed Mean of M-values. Analyses of DEGs were performed using the Bioconductor/R package edgeR v3.40.0. DEGs were selected based on an FDR of ≤ 0.05 with the adjusted $P$-value calculated using Benjamini–Hochberg correction and visualized with Bioinfokit 2.1.0 run on Python. The protein-coding genes among the DEGs were selected. The potential detection of transcripts derived from the Chst2 knockout gene was

excluded. Clustering analysis of DEGs were performed using ClustVis (https://biit.cs.ut.ee/clustvis/). Heatmap and volcano plot showing DEGs were created using Prism 7.0 (GraphPad Software). Gene Ontology (GO) term enrichment analysis was performed using the PANTHER knowledgebase (https://geneontology.org/; GO Ontology database Carbon & Mungall, 2018).

### Glycomic analysis

The spinal cord tissues were harvested at a length of 8-mm, centered on the site of injury, 7 d post-injury. All experimental procedures, including delipidation of spinal cord samples, chromatographic conditions, glycosidase treatments, desialylation reactions, and matrix-assisted laser desorption/ionization-time of flight mass spectrometry (MALDI-TOF MS), have been described previously (Yagi et al, 2005, 2022), with the exception of slight modifications for the purification of 2-aminopyridine (PA)-labeled glycans and separation of glycans on an anion-exchange column. Delipidated spinal cord lysates were dried by lyophilization. The *N*-glycans in the dried samples were digested by hydrazinolysis under the conditions described previously (Yagi et al, 2005, 2022). Oligosaccharides were released from the extracts by heating with 0.2 ml of anhydrous hydrazine at 100°C for 10 h in an evacuated sealed tube. On-column removal of the excess hydrazine, peptide materials, and detergents, and on-column *N*-acetylation were performed by using a carbon column (GL-Pak Carbograph, GL Sciences). The reducing ends of the *N*-glycans were labeled with PA. The PA-labeled glycan mixture was then purified on a cellulose column. The mixture of PA-*N*-glycan derivatives was separated by high-performance liquid chromatography on a Mono-Q column (Cytiva) at 30°C at a flow rate of 1.0 ml/min using solvent gradient before and after sialidase treatments. Solvent A was aqueous ammonia (pH 9.0), and the solvent B was 50 mM ammonium acetate solution (pH 9.0). The column was equilibrated with the solvent A. After injection of the sample, the concentration of the solvent B was increased to 20% with a linear gradient in 30 min. The desialylated PA-*N*-glycans were individually separated and sequentially identified on a Shim-pack HRC-octadecyl-silica (ODS) column (Shimadzu). The identification of the *N*-glycan structures was based on their elution positions on these columns in comparison with the PA-labeled glycans in the GALAXY database and existing high-performance liquid chromatography data for sulfated oligosaccharides (Yagi et al, 2005, 2022). The structures of PA-labeled glycans not registered in the GALAXY database were characterized by exoglycosidase treatments, desulfation reactions, and mass spectrometric techniques. The glucose unit values on the ODS column and the average mass calculated from the m/z values of $[M-H]^-$ ions for PA-oligosaccharides were determined.

### Statistical analysis

The data were analyzed using repeated measures two-way ANOVA with Tukey's range test, an ordinary two-way ANOVA with Bonferroni's test, or unpaired *t* test using Prism 7.0 (GraphPad Software). Differences were regarded as significant for $P < 0.05$.

## Data Availability

RNA-Seq data generated in the present study have been deposited in SRA (BioProject ID: PRJNA1268590). All data supporting the findings of this study are available from the corresponding author upon reasonable request.

## Supplementary Information

## Acknowledgements

We thank Narentuya, Zui Zhang, Foyez Tahmina, Yohei Kozaki, and Justin Lee for their technical assistance. This study was supported by grants-in-aid from the MEXT/JSPS (JP24590349, JP15K08265, and JP16KK0202 to K Uchimura, and JP20KK0371 to K Nishitsuji), the JST FOREST program (JPMJFR2255 to H Yagi), and in part by the Takeda Science Foundation (K Uchimura), the Suzuken Memorial Foundation (K Uchimura), and the Mizutani Foundation for Glycoscience (K Uchimura). This work was also supported by the joint research program of the J-GlycoNet collaborative network (25D001), which is accredited as a joint use/research center by the MEXT.

### Author Contributions

M Morozumi: formal analysis, investigation, and writing—original draft.
T Ozaki: formal analysis and investigation.
K Nishitsuji: data curation, software, formal analysis, funding acquisition, investigation, and writing—review and editing.
Y Takeda-Uchimura: investigation and writing—review and editing.
A Matsumoto: investigation.
S Ito: investigation.
S Imagama: resources.
N Ishiguro: resources.
H Yagi: formal analysis, funding acquisition, and investigation.
K Kato: resources.
TO Akama: resources.
T Kosugi: resources.
S Maruyama: resources.
K Kadomatsu: resources.
SD Rosen: resources and writing—review and editing.
LJ Noble-Haeusslein: resources, formal analysis, methodology, and writing—review and editing.
K Uchimura: conceptualization, resources, data curation, formal analysis, supervision, funding acquisition, validation, investigation, visualization, methodology, project administration, and writing—original draft, review, and editing.

### Conflict of Interest Statement

The authors declare that they have no conflict of interest.

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
