## [Reviewer comments · Life Science Alliance]

Life Science Alliance

Enhanced Locomotor Recovery in Mice Lacking GlcNAc6ST1 and GlcNAc6ST4 Following Spinal Cord Injury

Kenji Uchimura, Masayoshi Morozumi, Tomoya Ozaki, Akiyuki Matsumoto, Sadayuki Ito, Shiro Imagama, Naoki Ishiguro, Tomoya Akama, Kazuchika Nishitsuji, Steven Rosen, Linda Noble-Haeusslein, Yoshiko Takeda-Uchimura, Hirokazu Yagi, Koichi Kato, Tomoki Kosugi, Shoichi Maruyama, and Kenji Kadomatsu

DOI: <https://doi.org/10.26508/lsa.202503469>

Corresponding author(s): Kenji Uchimura, *Unité de Glycobiologie Structurale et Fonctionnelle*

Review Timeline:	Submission Date:	2025-07-26
	Editorial Decision:	2025-07-29
	Revision Received:	2025-08-19

Scientific Editor: *Tim Fessenden*

Transaction Report:

Please note that the manuscript was previously reviewed at another journal and the reports were taken into account in the decision-making process at *Life Science Alliance*. Since the original reviews are not subject to Life Science Alliance's transparent review process policy, the reports and author response cannot be published.

July 29, 2025

RE: Life Science Alliance Manuscript #LSA-2025-03469-T

Kenji Uchimura
UGSF CNRS
UGSF
Unknown
Univ. Lille
Lille F-59000
France

Dear Dr. Uchimura,

Thank you for submitting your manuscript entitled "Enhanced Locomotor Recovery in Mice Lacking GlcNAc6ST1 and GlcNAc6ST4 Following Spinal Cord Injury" to Life Science Alliance. Please see to the final revisions necessary to meet our formatting guidelines.

- We require only the manuscript file as an editable Word file. Please remove the PDF file.
- Please provide a clean manuscript file without track changes.
- It is recommended to exclude figures from the manuscript text and upload them separately.
- Please upload all figure files individually, including the supplementary figure files; all figure legends should only appear in the main manuscript file.
- Please add a Running Title and a Summary Blurb/Alternate Abstract in our system.
- Please add a Category and Keywords for your manuscript in our system.
- Please add the X and Bluesky handles of your host institute/organization, as well as your own and/or one of the authors in our system.
- Please be sure that the authorship listing and order are correct and match between the system and the manuscript file.
- Please consult our manuscript preparation guidelines <https://www.life-science-alliance.org/manuscript-prep> and make sure your manuscript sections are in the correct order.
- Please include a Data Availability section.
- Please add an Author Contributions section to the system.
- In the text of the manuscript, a reference should be cited by author and year of publication; 'et al' should be used if there are more than two authors (i.e., Smith & Jones, 2003; Smith et al, 2000). In-text citations of work posted on preprint servers must be clearly labeled as non-peer-reviewed work, e.g., (NAME et al., 2017; preprint) or (2; preprint).
- In the references section, citations should be listed with the authors' surnames and initials inverted. Where there are more than 10 authors on a paper, the first 10 will be listed, followed by 'et al.'.
- Please add your main, supplementary figure, and table legends to the main manuscript text after the references section.
- Please include supplementary references in the main references list.
- Please add callouts for Figure S4A-C to your main manuscript text.

A. FINAL FILES:

B. MANUSCRIPT ORGANIZATION AND FORMATTING:

Thank you for your attention to these final processing requirements.

Sincerely,

August 20, 2025

RE: Life Science Alliance Manuscript #LSA-2025-03469-TR

Dr. Kenji Uchimura
Unité de Glycobiologie Structurale et Fonctionnelle
UGSF
Building C9, Mendeleiev Ave
CNRS - UMR 8576 / University of Lille
Villeneuve d'Ascq 59655
France

Dear Dr. Uchimura,

Thank you for submitting your Research Article entitled "Enhanced Locomotor Recovery in Mice Lacking GlcNAc6ST1 and GlcNAc6ST4 Following Spinal Cord Injury". It is a pleasure to let you know that your manuscript is now accepted for publication in Life Science Alliance. Congratulations on this interesting work.

DISTRIBUTION OF MATERIALS:

Again, congratulations on a very nice paper. I hope you found the review process to be constructive and are pleased with how the manuscript was handled editorially. We look forward to future exciting submissions from your lab.

Sincerely,
